# A Systematic Review of Inverse Agonism at Adrenoceptor Subtypes

**DOI:** 10.3390/cells9091923

**Published:** 2020-08-19

**Authors:** Martin C. Michel, Martina B. Michel-Reher, Peter Hein

**Affiliations:** 1Department of Pharmacology, Johannes Gutenberg University, 55131 Mainz, Germany; michelma@uni-mainz.de; 2TherapeutAix UG, 52066 Aachen, Germany; pehein@gmail.com

**Keywords:** adrenoceptor, constitutive activity, drug development, inverse agonism

## Abstract

As many, if not most, ligands at G protein-coupled receptor antagonists are inverse agonists, we systematically reviewed inverse agonism at the nine adrenoceptor subtypes. Except for β_3_-adrenoceptors, inverse agonism has been reported for each of the adrenoceptor subtypes, most often for β_2_-adrenoceptors, including endogenously expressed receptors in human tissues. As with other receptors, the detection and degree of inverse agonism depend on the cells and tissues under investigation, i.e., they are greatest when the model has a high intrinsic tone/constitutive activity for the response being studied. Accordingly, they may differ between parts of a tissue, for instance, atria vs. ventricles of the heart, and within a cell type, between cellular responses. The basal tone of endogenously expressed receptors is often low, leading to less consistent detection and a lesser extent of observed inverse agonism. Extent inverse agonism depends on specific molecular properties of a compound, but inverse agonism appears to be more common in certain chemical classes. While inverse agonism is a fascinating facet in attempts to mechanistically understand observed drug effects, we are skeptical whether an a priori definition of the extent of inverse agonism in the target product profile of a developmental candidate is a meaningful option in drug discovery and development.

## 1. Introduction

Adrenoceptors (AR) mediate many of the physiological and pathophysiological functions of the neurotransmitter noradrenaline and the adrenal hormone adrenaline, for instance, in the heart, airways, liver, and urogenital tract. Three families of AR exist, consisting of α_1_-AR, α_2_-AR, and β-AR, and include three subtypes each (α_1A_, α_1B_, α_1D_, α_2A_, α_2B_, α_2C_, β_1_, β_2_, and β_3_) [1,2,3]. Early receptor theory assumed that ligands can interact with receptors by either activating them (agonists) or preventing activation by other ligands (antagonists). This was initially amended by findings that even high concentrations of some ligands produce smaller responses than reference agonists, i.e., are partial agonists. Largely driven by experiments with a heterologous expression of high receptor densities and/or constitutively active mutants (CAM) of receptors [4], it then emerged that some compounds originally assumed to only act by inhibiting effects of agonists (conventionally called ‘antagonists’) can also reduce receptor activity in the absence of agonists; this property was described by the term inverse agonism (IA) [5,6,7,8]. While originally described for GABA_A_ receptors, it is now clear that IA can occur at most, if not all, G protein-coupled receptors (GPCRs), including angiotensin receptors [9], muscarinic acetylcholine receptors [10], histamine receptors [11], and dopamine receptors [12]. While early studies proposed that IA may be the exception among ligands, the opposite appears to be true, i.e., that very few ligands do not change receptor activity (i.e., are neutral or silent antagonists) and most compounds originally classified as antagonists either cause a small extent of receptor activation (weak partial agonists) or act as inverse agonists. Therefore, it has been estimated that 85% of all compounds previously classified as antagonists are inverse agonists [13]. Of note, measured IA is only partly an intrinsic property of a compound, and also depends on the model in which it is investigated; thus, a compound may exhibit partial agonism in some models, neutral antagonism in others, and IA in even further models [14,15]—most likely depending on the tone/constitutive activity of a system (see below). Evidence from receptor crystal structures has indicated that each ligand induces a unique conformation of a receptor and that conformations induced by inverse agonists differ from those induced by neutral antagonists (see Section 6).

This article will briefly discuss methodological aspects of studying IA (Section 3), which will be followed by a systematic description of compounds exhibiting IA at AR subtypes (Section 4) and how their effects on complex biological systems, such as living animals or humans, may deviate from those of neutral antagonists (Section 5). Finally, we will discuss the molecular basis of IA and its impact on drug development and the treatment of disease (Section 7).

## 2. Search Strategy

We searched PubMed using the term “inverse agonist” in combination with any of the terms “alpha-1”, “alpha-2”, “beta-1”, “beta-2”, or “beta-3”. Two authors (M.B.M.-R. and M.C.M.) removed duplicates retrieved by more than one search (*n* = 40). The title and abstract of retrieved references were independently screened by two authors (M.C.M. and P.H.) and grouped as an original paper, as a review article, or as obviously out of scope; the latter included those not dealing with IA at AR but, for instance, with α1 subunits of GABA receptors. Retracted papers were also excluded from the analyses (*n* = 1). Full texts were retrieved for original and review papers potentially considered to be in scope and those where the two examiners did not agree on “obviously out of scope” based on the title and abstract. The reference list of the review articles was manually searched for additional applicable original studies. A PRISMA flow chart of our search results is shown as Figure 1.

## 3. Methodological Aspects of Studying IA

IA can, in principle, be assessed by any functional response. Frequently applied assays include GTPγS binding as a very proximal assay of receptor activity [16,17,18]; more distally, early second messengers, such as the formation of inositol phosphates (IP) [19], formation of cAMP [15,20], or inhibition of the formation of cAMP [21,22]; or, even more distally, up-regulation of the receptor number [23,24,25,26,27]. Less frequently applied readouts involve AR phosphorylation [26,28], FRET signals for receptor/G-protein interactions [29,30,31], GTPase activity [32], β-arrestin recruitment [33], the activity of phospholipase D [23] or extracellular signal-regulated kinase [14,34], free intracellular Ca^2+^ concentrations [35], the modulation of ion channels such as L-type Ca^2+^ channels [36] or the iK_Ca_1 K^+^ channel [37], or the modulation of cAMP gene transcription by acting on cAMP response elements [15,38]. However, it needs to be considered that the more distant an observed response occurs in the signal transduction cascade, the greater the biological complexity of the assay and the possibility that it becomes affected by factors other than IA [39]. α_1_-, α_2_-, and β-ARs use distinct canonical signaling pathways, i.e., involving G_q/11_, G_i/o_, and G_s_, respectively; accordingly, studies of IA have largely applied IP formation, GTPγS binding, and cAMP formation, respectively, reflecting this differential G protein coupling.

Similar to agonism, IA can occur at orthosteric and allosteric sites [40,41], and similar to classic antagonism, IA can present as competitive or non-competitive [42] and can exhibit stereo-specificity [43]. Moreover, AR ligands with IA are not necessarily small molecules, but can also be antibodies [44].

It appears obvious that a reduction in the basal state is easiest to detect if the tone/constitutive activity of the system is high (better signal/noise ratio). The basal state depends on endogenous features of the model under investigation, e.g., the expression of various molecules involved in the signal transduction chain up to the point being measured. This can be experimentally manipulated, for instance, by over-expressing receptors [42,45,46,47,48]; by co-expression with a G protein [18,49,50] or an adenylyl cyclase isoform [51]; by studying receptor/G protein fusion proteins [32,49,52]; by using constitutively active mutants (CAM) of the receptor [17,18,19], and by sensitizing the signaling system, as has been done in cells expressing opioid receptors by pre-treatment with morphine [20,53]. Of note, CAM can include naturally occurring receptor gene polymorphisms [30,54,55], which frequently occur [56].

Considering this, it is clear that IA is not an intrinsic property of a ligand, but is context-dependent. Therefore, a given compound may behave as an inverse agonist in some models, as a neutral antagonist in others, and as a (weak) partial agonist in even further models [57]. It may even exhibit these distinct properties for one vs. another response within a model, particularly if one read-out has a greater basal tone (signal/noise ratio) than the other [14]. This has two implications: Firstly, the degree of IA may differ between ligands for a given read-out (see, for instance, Section 4.3). Second, the degree of IA may differ between read-outs for a given ligand; for instance, the degree of IA, as measured in right and left atria and the ventricle of a rat heart, that considerably differs for any given compound [58]. Accordingly, ligands may exhibit partial IA [17,23,57]. While ligands from some chemical classes may exhibit stronger IA than others, it is typically not limited to one chemical class of ligands [19,23,57] and can even occur at AR with peptide ligands [40]. Therefore, the term inverse agonist should only be used specific to a context, similar to the term partial agonist [59].

The above can be conceptualized by imagining the receptor activity as an “output” on a continuum ranging from “no signaling” to “maximal signaling” (Figure 2). The level of basal activity, i.e., in the absence of a ligand, is determined by intrinsic receptor properties (e.g., activating mutations) and its local environment (e.g., ions, G protein availability, temperature, and localization to signaling domains). A specific ligand is then simply one additional “input”, which, together with all other inputs, determines the signaling output level. For example, a ligand that would set the output level to 20% of the maximal effect can be considered a partial agonist if the other inputs set the receptor at a signaling level below 20%, an inverse agonist if the other inputs set a level of higher than 20%, and a neutral antagonist if all other inputs have already set the signaling output to 20%. Changing the input, e.g., by introducing a CAM that causes the activity level to be 40%, subsequently changes the apparent classifications for specific ligands; in this example, a ligand behaving as a “neutral antagonist” at 20% appears to be an “inverse agonist” at 40%. In this framework, the terms “inverse agonist”, “neutral agonist”, and “partial agonist” begin to merge, since the traditional method used to classify a ligand (measuring changes in the signaling output) can give different results, depending on the dynamic nature of the receptor activity level in the absence of a ligand. Similar concepts have previously been proposed, e.g., to consider efficacy as a vector [13]. To complicate things further, given inputs can lead to different output levels as different signaling endpoints are considered.

Stringent proof of IA requires that the inverse effect of one ligand is blocked by a neutral antagonist; however, most studies in the field have assumed that a given compound shown once or more often to reduce the basal receptor output agonist consistently acts via IA, but this is not necessarily the case. For instance, ICI 118,551 has repeatedly been shown to be an inverse agonist at β_2_-AR (see below); however, its inhibitory effect in the rat endothelium was neither mimicked by other known inverse agonists (e.g., carvedilol or nadolol) nor blocked or reversed by antagonists (e.g., propranolol) or agonists (e.g., salbutamol) [60], indicating that it may have acted by β-AR-independent mechanisms in this model. While this may be the exception, IA as an explanation of an observed effect should not be assumed too easily.

## 4. Compounds Exhibiting IA at AR Subtypes in Cellular Models

The subsequent sections are primarily ordered by subtypes within a subfamily of α_1_-, α_2_-, and β-AR. Most research has been reported for isolated cells, very often transfected with the receptor of interest, but in some cases also with endogenously expressed receptors.

### 4.1. α_1_-AR

The initial literature on IA at α_1_-AR has previously been summarized in a narrative review [61]. IA at α_1_-AR has largely been studied using cell lines such as COS-7 [19,45], CHO [16], or HEK cells [17,62,63] or rat-1 fibroblasts [23,24,34,35,57,64] transfected with cloned wild-type (WT) receptors or, in some cases, CAM thereof [16,19,23,45,57,63]. One study of an isolated rat uterine cervix reported a pertussis toxin-sensitive stimulation of GTPγS binding by WB 4101, whereas phenylephrine had the opposite effect; the concentration-dependent effects of WB 4101 on cervical tone were inhibited by phentolamine [65]. However, the α_1_-AR subtype involved in this effect has not been determined.

#### 4.1.1. α_1A_-AR

The broadest evaluation of IA at α_1A_-AR characterized 24 compounds representing various chemical families, including *N*-arylpiperazines, 1,4-dihydropyridines, imidazolines, benzodioxanes, phenylalkylamines, and quinazolines, for their ability to inhibit basal IP formation in COS-7 cells transfected with CAM [19] (Figure 3). Among the *N*-arylpiperazines, some compounds (WAY 100365) exhibited strong IA, some (5-methylurapidil and BMY 7378) displayed moderate IA, and others (REC 15/2739, REC15/2869, REC 157/3011, and REC 15/3039) did not show IA. All tested compounds from the other chemical classes, including the clinically used alfuzosin, phentolamine, prazosin, spiperone, tamsulosin, and terazosin, exhibited moderate to strong IA. Nine of these compounds were also tested at WT α_1A_-AR with largely similar results, but the extent of IA appeared greater than with the CAM receptor for 5-methylurapidil, BE 2253, and REC 15/2869.

Other investigators tested smaller panels of ligands, but evaluated them in multiple assays at the CAM of human α_1A_-AR transfected into CHO cells [16]. They confirmed IA as measured for IP formation for BMY 7378, phentolamine, prazosin, and WB 4101, and extended this to HV723, whereas silodosin (formerly known as KMD-3213) did not exhibit IA. As a second indicator of IA, they tested the ability to cause receptor up-regulation; this yielded qualitatively similar results to the IP formation assay, but the extent of IA appeared stronger in the IP than in the up-regulation assay for BMY 7378 and WB 4101, where the opposite was observed for HV 723 and phentolamine. Neither prazosin nor silodosin exhibited IA, as assessed by IP formation against WT α_1A_-AR. In a third assay, prazosin but not silodosin inhibited GTPγS binding. Finally, silodosin attenuated the inhibitory effects of prazosin in the IP and GTPγS assay, strengthening the evidence for IA as an underlying mechanism due to reversal by a neutral antagonist.

In contrast to the above two studies, others reported a lack of IA of phentolamine and prazosin against basal IP formation with WT human α_1A_-AR transfected into Rat-1 fibroblasts under conditions where the two compounds blocked the effect of phenylephrine [24]. Nonetheless, prazosin increased the number of α_1A_-AR expressed at the cell surface, irrespective of the presence of the agonist phenylephrine. The lipid raft disrupting agent methyl-β-cyclodextrin blocked the receptor up-regulation by prazosin. Methyl-β-cyclodextrin decreased the affinity of phenylephrine, but increased that of prazosin or phentolamine. While other investigators have reported minor differences in affinity between WT and CAM for some, but not other, inverse agonists, a systematic pattern was not evident [16,19].

Neither phenylephrine nor prazosin affected continuous internalization rates of human α_1A_-AR transfected into Rat-1 fibroblasts [64]. Similarly, neither adrenaline nor prazosin affected dimerization between human α_1A_-AR and hamster α_1B_-AR transfected into HEK cells [62]. In conclusion, most, but not all, antagonists exhibited IA at α_1A_-AR, but the extent of IA differed considerably between compounds. Some evidence indicates that a given ligand may exhibit a greater degree of IA for a certain assay (e.g., IP formation vs. receptor up-regulation), whereas other ligands may exhibit the opposite preference within the same study. Therefore, the possibility exists that the IA may concomitantly involve a component of biased agonism.

#### 4.1.2. α_1B_-AR

The initial report on IA at α_1B_-AR was based on a CAM of the hamster receptor and showed the inhibition of IP formation by prazosin and phentolamine [45]. In a follow-up study, the same group tested 24 antagonists from different chemical classes for their ability to inhibit IP formation with both WT and CAM of the human α_1B_-AR [19]. Most ligands exhibited IA against the CAM and fewer against the WT. Generally, the extent of IA appeared to be larger than for that at α_1A_-AR investigated within the same study (see above). There was no systematic difference in affinity between CAM and the WT receptor. Using cells obtained from these investigators, other confirmed the IA of a range of compounds from various chemical classes against IP formation at the CAM [57]. Interestingly, the tested quinazolines alfuzosin, doxazosin, prazosin, and terazosin exhibited a similar level of IA, whereas that of the non-quinazoline BE 2254 was considerably weaker. SB 216,469 behaved as a neutral antagonist and tamsulosin even behaved as a weak partial (not inverse) agonist; SB 216,469 inhibited both the IA by quinazolines and the partial agonism of tamsulosin. None of the tested ligands exhibited IA at the WT receptor. An independent group also did not detect IA of prazosin for IP formation at the WT human α_1B_-AR and observed weak partial agonism for the activation of extracellular signal-regulated kinase [34].

Using phospholipase D activity as the read-out with a CAM α_1B_-AR receptor, IA was reported by various ligands, but 5-methyl-urapidil had a somewhat, and tamsulosin a considerably, weaker efficacy than the other tested compounds [23]. As part of the same study, up-regulation of the CAM α_1B_-AR was observed by the same ligands and again, tamsulosin and 5-methyl-urapidil showed a weaker IA efficacy. Up-regulation of the CAM α_1B_-AR was also reported by others upon exposure to various inverse agonists, but none of them caused an intracellular redistribution [63]. Neither the full agonist adrenaline nor the inverse agonist prazosin affected dimerization between the α_1A_- and α_1B_-AR [62]. Using a fusion protein of WT or CAM of the hamster α_1B_-AR and the α-subunit of G_11_, others detected IA based on the inhibition of GTPγS binding with a rank order of efficacy of phentolamine > WB4101 > corynanthine > HV723 > urapidil [17].

#### 4.1.3. α_1D_-AR

IA has been studied to a lesser extent at the α_1D_-AR subtype compared to the other α_1_-AR subtypes. 5-methyl-urapidil, BMY 7378, chloroethylclonidine, and phentolamine reduced basal free intracellular Ca^2+^ concentrations in Rat-1 fibroblasts transfected with α_1D_-AR, whereas WB 4101 only had very small effects; however, WB 4101 inhibited the effects of the inverse agonists and noradrenaline [35]. Other investigators reported that prazosin reduced IP formation and the activity of extracellular signal-regulated kinase in Rat-1 fibroblasts transfected with human α_1D_-AR, whereas this had not been observed with α_1B_-AR within the same study [34]. They also observed an intracellular redistribution of the α_1D_-AR upon exposure to prazosin.

In conclusion, the majority of all tested compounds previously considered as antagonists exhibited IA at cloned α_1_-AR subtypes (Figure 3). This was observed with subtypes from multiple species (e.g., hamsters and humans) and in a range of expression systems, including COS-7, CHO, HEK, and Rat-1 cells, indicating that it is a potentially universal phenomenon. While most studies focused on IP formation, IA has also been demonstrated for a range of other signaling pathways and for more distal responses, such as receptor up-regulation. IA was detected more consistently with CAM than with WT of the receptor, confirming that the detection of IA depends on the extent of basal activity of the functional response under investigation. Similar to agonists, inverse agonists exhibited a range of efficacies, with a general trend for qinazolines, including several clinically used drugs (alfuzosin, doxazosin, prazosin, and terazosin), typically having the greatest efficacy as inverse agonists (Figure 3). The clinically used non-quinazoline tamsulosin often (but not always) exhibited weaker IA and in some cases, where quinazolines showed IA, behaved as a weak partial agonist; whether this contributes to observed differences in clinical profiles of these ligands remains unclear. Only a few compounds were close to being neutral antagonists; these similarly inhibited responses to agonists and inverse agonists.

### 4.2. α_2_-AR

IA at α_2_-AR has also been studied in a variety of transfected cell lines, including COS-7 [50], CHO [18,21,22,66,67], HEK [29,68], HEL [43], and PC-12 cells [46,69,70,71], mostly using WT, but in some cases also a CAM of the receptor [18,22,29,66,67,71] or co-transfection with a G_o_ protein [18,66]. In contrast to the studies with α_1_-AR, those with α_2_-AR have at least in some cases been performed with a natively expressed receptor, e.g., in C6 glioma [21], NG 108-15 neuroblastoma [72], and HepG2 hepatocarcinoma cells [73].

A study using saturation and competition radioligand binding studies in rat brain sections (mix of α_2_-AR subtypes) was based on the premise that the presence of GTP decreases the affinity of agonists and increases that of inverse agonists when tested in the presence of magnesium [74]. GTP increased the affinity of RX 821002 and decreased that of rauwolscine with inconclusive data for MK-912. These data are difficult to interpret because earlier studies comparing the effects of multiple buffers in the rat cerebral cortex had not reported the effects of GTP on the affinity of either RX 821002 or rauwolscine [75].

#### 4.2.1. α_2A_-AR

Using WT-transfected PC12 cells, the inhibition of GTPγS binding was reported for rauwolscine, yohimbine, phentolamine, idazoxan, and WB4101 [69] and in a later report by tolazoline and some of its analogs [70]. In a follow-up study using not only the WT, but also a CAM receptor, these investigators focused on the molecular mechanisms underlying the IA by rauwolscine [71]. Treatment with protein kinase C inhibitors such as bisindolyl-maleimide, calphostin C, chelerythrine, or staurosporin, but not those of several other protein kinases, almost abolished the inhibitory effect of rauwolscine on GTPγS binding. The IA of rauwolscine was also abolished in a Ca^2+^-free medium. While a G_i1/2_ antiserum had stronger inhibitory effects on adrenaline-stimulated GTPγS binding than a G_s_ antiserum, only the latter reduced basal GTPγS binding, indicating that different G proteins are involved in constitutive as compared to agonist-stimulated activity.

Using CHO cells transfected with the human WT or CAM receptor and an additional rat G_o_ protein, reduced basal GTPγS binding was found for RX 811059 and its (+)-enantiomer, (+)-RX 821002, RS 15385, and yohimbine, whereas fluparoxan and WB4101 exhibited partial IA and atipamezole and dexefaroxan were neutral antagonists; atipamezole inhibited the agonism by UK 14,304 and the IA by (+)-RX 811059, with similar pK_B_ values [18]. A follow-up study from the same group confirmed the IA of (+)-RX 811059 and neutral antagonism of atipamezole for GTPγS binding [66]. Surprisingly, a 48 h incubation period with the inverse agonist (+)-RX 811059, the neutral antagonist atipamezole, and the efficacious agonist medetomidine similarly increased α_2A_-AR protein expression within that study, both for WT and the CAM receptor. These findings challenge the assumption that up-regulation of the target receptor can easily be interpreted as a result of IA; rather, they support the idea that structural stabilization of the receptor may be involved in up-regulation, irrespective of the nature of the ligand [66]. In another follow-up study, this group tested various analogs of dexefaroxan for IA to inhibit of GTPγS binding in transfected CHO for cAMP formation in transfected C6 glioma cells [21]. While chemically closely related, the efficacy for GTPγS binding differed widely between the analogs and included inverse agonists (RX 851062), neutral antagonists (RX 851057), partial agonists (RX 821008), and full agonists (RX 821010); however, none of these compounds exhibited positive or negative efficacy in the cAMP assay, further demonstrating that the presence of IA depends on both the intrinsic properties of a compound and those of the assay system.

Two groups of investigators have tested known inverse agonists for their effects on receptor/G protein interactions using FRET-based approaches in transfected HEK cells. IA could be detected when conformational changes were examined using intramolecular FRET (Vilardaga et al., 2005). When looking at FRET between α_2A_-AR/M_4_ muscarinic receptor dimers and G proteins, the agonist showed a decrease, while no effect was seen with inverse agonists; this was interpreted as evidence that the receptor dimer and the G protein heterotrimer exist at the cell membrane in the resting state in a pentameric complex (Nobles et al., 2005).

In studies with CHO cells transfected with either WT or the CAM receptor, several ligands exhibited IA with a rank order of inverse efficacy for the modulation of forskolin-stimulated cAMP formation of rauwolscine > yohimbine > RX821002 > MK912, whereas phentolamine and idazoxan were largely neutral antagonists; the irreversible ligand phenoxybenzamine also did not have an effect [22]. IA based on cAMP formation has also been demonstrated for levomedetomidine, idazoxan, rauwolscine, and atipamezole with endogenously expressed α_2A_-AR in HEL cells [43]. In contrast to these findings, others using C6 glioma cells did not exhibit the effects of various RX821002 analogs for the modulation of cAMP formation [21].

While the stimulation of phospholipase C is not a typical signaling response of α_2A_-AR [1], such coupling was observed in Cos-7 cells transfected with WT and a CAM α_2A_-AR; IP formation by the CAM (but not the WT) was further enhanced upon co-transfection with the α-subunit of murine G_15_ [50]. However, none of the ligands shown by the same group to exhibit IA for GTPγS binding [18,21,66], including MK 912, WB 4101, RS 15385, RX 811059, and RX 821002, exhibited IA in this model; compounds that were neutral antagonists for GTPγS binding, such as dexefaroxan, idazoxan, atipamezole, BRL 44408, and SKF 86466, exhibited partial agonism for IP formation. A phospholipase C-independent elevation of intracellular Ca^2+^ concentrations could be observed in HEL cells [76]. In these cells, Ca^2+^ levels were increased by dexmedetomidine and lowered by levomedetomidine, idazoxan, and rauwolscine [43]. The neutral antagonist MPV-2088 was inhibited by responses. Despite exhibiting IA in both the Ca^2+^ and cAMP assay in HEL cells, levomedetomidine behaved as a partial agonist in rat vas deferens.

#### 4.2.2. α_2B_-AR

Yohimbine was shown to exhibit IA for GTPγS binding at α_2B_-AR endogenously expressed in NG 108-15 neuroblastoma cells [72]. A lowering of intracellular Ca^2+^ concentrations was observed with chlorpromazine and rauwolscine RX821002, but not with ARC 239, MK 912, or phentolamine, in PC12 cells transfected with α_2B_-AR, whereas atipamezole was a partial agonist [46]. However, very high (3.8 pmol/mg protein) receptor expression was required to detect both IA and partial agonism, and this was not seen with lower, but still high, expression levels (1.3 pmol/mg protein). The same group also reported on WT and CAM α_2B_-AR expressed in CHO cells [67]. In line with their data from the NG 108-15 cells, RX 821002 exhibited a lowering of the intracellular Ca^2+^ concentrations with the CAM, but not the WT, receptor.

#### 4.2.3. α_2C_-AR

Our search identified only one study related to IA by α_2C_-AR: Treatment with RX821002 or yohimbine increased the receptor number, as assessed by radioligand binding in HepG2 hepatocarcinoma cells endogenously expressing the receptor, whereas treatment with UK 14,304 reduced it and phentolamine did not display an effect [73]. The regulation of α_2C_-AR binding sites was not accompanied by changes in corresponding mRNA levels, but was rather a consequence of increased receptor degradation by the agonist and decreased degradation by the inverse agonist.

In conclusion, knowledge on IA at α_2_-AR is largely driven by that on α_2A_-AR. While demonstrations of IA tested as reductions of GTPγS binding are very consistent across studies and investigations, the effects on the canonical pathways of inhibition of cAMP formation are less consistent. Studies on other readouts are either contradictory or too limited in number to reach robust conclusions. While the detection of IA was facilitated by a higher expression of WT, CAM, or the co-expression of G protein α-subunits, it has also been reported with multiple cell lines endogenously expressing the receptor.

### 4.3. β-AR

Subtypes of β-AR have been studied more extensively for IA than those of α_1_- and α_2_-AR combined. This is mostly due to studies with β_2_-AR, and most likely because the β_2_-AR was the first cloned G protein-coupled receptor [77] and became a general paradigm for studying GPCRs. In contrast, our search did not identify any studies related to IA at β_3_-AR. β-ARs are the only AR subfamily for which IA in tissues has been explored in many studies (see Section 5).

#### 4.3.1. β_1_-AR

IA at β_1_-AR has been explored using transfected cell lines; this was mostly done in HEK cells, although CHO [78] and CHW cells were also used [54]. They were mostly transfected with the human receptor, but in some cases, also with the turkey receptor [78,79]. Transfections were mostly conducted with the WT β_1_-AR, but often also with CAM [80] and/or naturally occurring variants of the receptor [30,54,55]. To a limited extent, IA has been studied at the signal transduction level in cells endogenously expressing the β_1_-AR [81].

Most investigators have used cAMP formation as a readout for IA. The inhibition of basal cAMP formation was consistently observed with metoprolol [54,55,80]. IA has also been reported for CGP 20,712 [54], whereas the detection of IA by propranolol with the human receptor [14] was not confirmed with the turkey ortholog [79]. Two studies from the same group also did not observe IA by ICI 118,551 at the turkey β_1_-AR [78,79]. Bucindolol at the human receptor [14] and carazolol at the turkey receptor [78] were reported to be partial agonists, with the latter being noteworthy because it has consistently been shown to exhibit IA at the β_2_-AR (see below). The lowering of basal cAMP formation has also been found for CGP 20,712 in the rat anterior pituitary gland, representing a bona fide β_1_-AR, while this was abolished by carvedilol or pre-treatment with pertussis toxin, and no IA was observed for betaxolol or propranolol in this model [81].

Only a few studies have tested IA for signaling responses other than cAMP formation, all at human β_1_-AR transfected into HEK cells and sometimes in direct comparison to ligand effects on cAMP. Using a FRET assay to directly monitor the receptor/G protein-interaction, carvedilol and, to a lesser extent, bisoprolol and metoprolol were found to exhibit IA [30]; the IA displayed by carvedilol was stronger in the naturally occurring (hypoactive) Arg^389^ than in the Gly^389^ variant. Using a different FRET probe that monitors cAMP levels, carvedilol also exhibited stronger IA with the Arg^389^ than the Gly^389^ variant, with bisoprolol and metoprolol not displaying measurable IA in that assay. In a comparison of cAMP formation and the activation of extracellular signal-regulated kinase, propranolol was an inverse agonist, whereas bucindolol was a partial agonist, for cAMP, whereas both were partial agonists for kinase activation [14]. Metoprolol exhibited stronger IA for cAMP formation at CAM R384E and R384Q mutations of the β_1_-AR compared to WT; in contrast to the WT, the mutated receptor largely exhibited intracellular localization and was redistributed to the plasma membrane in the presence of metoprolol or CGP 20,712 [80].

#### 4.3.2. β_2_-AR

IA has been studied more often at β_2_-AR than any other AR subtype. Accordingly, this has been conducted in a wide range of models, including transfected mammalian COS-7 [47,53], BC3H1 [42], CHW [49,54], CHO [15,28,48,82,83], HEK [26,27,31,33,84,85], NG 108-15 cells [25,51,53,86], Burkitt lymphoma [38], H9C2 cells [87], fibroblasts [84], and insect Sf9 cells [32,48,52,88]. Models with an endogenous expression of β_2_-AR have also been used, including A431 [20] cells and cardiomyocytes [36,89,90]. While the most frequently used readout was cAMP formation, others included FRET assays for the receptor/G protein interaction [31,91], G protein activity [32], arrestin recruitment [33], receptor phosphorylation [26], intracellular Ca^2+^ levels [90], Ca^2+^ channel activity [36], the activation of extracellular signal-regulated kinase [15], reporter gene assays [15,38], and receptor up-regulation [25,26,27,85,86].

Several approaches have been applied to explore IA at the level of the receptor/G protein interaction. ICI 118,551 exhibited IA in a G protein activation assay [91]. FRET sensors responsive to the β_2_-AR/G protein interaction showed IA for ICI 118,551 and, to a lesser extent, for metoprolol [31]. Others reported that ICI 118,551 inhibited the phosphorylation of a CAM β_2_-AR, whereas propranolol had no effect and isoprenaline increased phosphorylation; phosphorylation of the WT β_2_-AR was also stimulated by isoprenaline, but not affected by either ICI 118,551 or propranolol [28]. ICI 118,551 also inhibited GTP hydrolysis of a constitutively active fusion protein between the β_2_-AR and the α-subunit of G_s_ [32]. Extending these observations, acute exposure to nadolol (5 min) reduced cAMP formation and forskolin-stimulated phosphorylation of the β_2_-AR at Ser^355^ and Ser^356^, whereas longer exposure (24 h) increased cAMP formation, presumably by up-regulation of the receptor, and did not change receptor phosphorylation [26]. In contrast to the agonist adrenaline, ICI 118,551 did not affect arrestin recruitment [33].

More than 20 studies have demonstrated the inhibition of basal cAMP formation via WT β_2_-AR: These studies demonstrated IA for ICI 118,551 as the most frequently studied compound [15,20,28,31,32,33,42,49,51,52,53,54,83,84,85,88,89,90], but also for alprenolol [42,48,82], atenolol [15,42], betaxolol [28,51,82,85,86], bisoprolol [15], cyanopindolol [88], dichloroisoproterenol [48], labetolol [48,82], metoprolol [15,31], nadolol [26], pindolol [48,82], propranolol [15,42,48,82,88], sotalol [15,86], and timolol [15,53] [48,51,82,84,86]. However, the IA exhibited by some of these compounds was not confirmed or found under other experimental conditions, for instance, for alprenolol [28,51,85], dichloroisoproterenol [82], labetolol [85], or propranolol [51,86]. This may reflect a lack of robustness of findings with a given compound, particularly if it exhibited only partial IA in the ‘positive’ studies or if the ‘positive’ studies were performed in models with a greater basal tone of the system than the ‘negative’ studies.

The presence of true IA was confirmed by the antagonism of reduced cAMP formation in the presence of ligands with considerably weaker IA or neutral antagonists [20,42,48,53,86]. As with other AR subtypes, the relative efficacy of IA differed considerably between ligands [15,48,82] (Figure 4).

The stimulation of cardiac β-AR can lead to elevations of intracellular Ca^2+^. A study of most cardiomyocytes found that concentrations of ICI 118,551 can reduce the elevated Ca^2+^ concentrations induced by the PI-3 kinase inhibitor LY 294002 or the phosphodiesterase type 4 inhibitor milrinone [90]. Similarly, the activation of L-type Ca^2+^ channels by a peptide corresponding to the second loop of the human β_2_-AR was inhibited by ICI 118,551, but not by alprenolol, in guinea pig cardiomyocytes [36]. While mostly occurring via β_1_-AR, the authors attributed these effects to β_2_-AR because of the low concentrations of ICI 118,551 being used and the high β_2_-selectivity of this compound.

Studies of transfected CHO cells found that several ligands exhibited IA for the activation of a reporter gene construct based on the cAMP response element; while the pattern was similar for various compounds for IA of cAMP formation and activation of the reporter gene construct, it was generally higher for the latter. Accordingly, it was detected for cAMP formation, but not the reporter gene construct, for some compounds (Figure 4) [15]. While activation of the reporter gene is a more downstream response than cAMP formation, it should be noted that according to that study, cAMP formation apparently occurred via a G protein-independent pathway. ICI 118,551 also exhibited IA in a similar assay in a Burkitt lymphoma cell line, whereas propranolol was a partial agonist in this model [38].

Several studies have used β_2_-AR upregulation to study IA. This was observed with betaxolol in NG 108-15 cells transfected with a CAM and to a lesser extent, when transfected with a WT receptor, whereas this was not seen with alprenolol with either [25]. Interestingly, the half-maximal concentrations of betaxolol for receptor upregulation were very similar to those of reducing basal adenylyl cyclase activity. Moreover, the upregulation of the receptor was not accompanied by a change of the G_s_ protein or mRNA, and depended on de novo protein synthesis. Additional studies from the same group confirmed the upregulation and demonstrated that it could be prevented by the neutral antagonists dihydroalprenolol and propranolol [86]. Upregulation of the β_2_-AR was also shown in transfected HEK cells for betaxolol, diyhydroalprenolol, ICI 118,551, or labetalol [85]; for betaxolol [27]; and for nadolol [26]. In H9C2 cells, ICI 118,551 was used based on its consistently shown IA as a probe for the presence of constitutive activity for the formation of nanoscale clusters, but had no effect in that model [87].

## 5. Effects of Compounds with IA Data for Tissue and In Vivo Function

Studies with isolated tissues and in vivo are particularly important in two ways: Firstly, they allow researchers to study tissue or whole organism consequences of IA, and secondly, by exploring responses with direct physiological relevance, they are more likely to be representative for effects related to a therapeutic situation if endogenously expressed receptors are studied. However, a key limitation of most of these studies is that employed tools typically allow researchers to assign IA to α_1_-, α_2_-, or β-ARs as a subfamily, but less so to directly link it to a specific subtype within a subfamily. Another limitation largely applicable to the in vivo studies is the difficulty of understanding whether a given response opposite to that of an agonist can be attributed to IA or alternatively explained by classic antagonism of endogenously present agonists [92,93]; this applies even more to the interpretation of human in vivo studies [94] in which much less experimental modulation is possible than in research animals for ethical reasons. One approach for addressing this is intra-study comparisons with ligands that were reported to be neutral antagonists.

### 5.1. α_1_-AR

One of the earliest reports on IA at AR in a complex physiological system was based on indirectly measuring the depletion of intracellular Ca^2+^ stores in rat aorta strips [39] and this highlights the complexity of studying IA in native tissue. The investigators initially contracted rat aortic strips with noradrenaline and then repeated noradrenaline exposure in Ca^2+^-free medium to deplete internal Ca^2+^ stores; after refilling of the intracellular stores, spontaneous increases in resting tone were observed, which were inhibited by antagonists such as benoxathian and WB 4101 in the absence of agonists.

In an isolated rat cervix, WB 4101 concentration-dependently increased GTPγS binding and increased tone, with both being modified by the day of pregnancy [65]. Phentolamine inhibited the WB 4101 effects on cervical tone, whereas other known inverse agonists, such as AH 11110A and BMX 7378, did not mimic the effect of WB4101. Others reported that WB 4101 and two of its analogs K^+^-induced contractions in the guinea pig ileum and left atrium, but not in the aorta, whereas various Ca^2+^-channel inhibitors exhibited comparable inhibition in all three tissues [95]; unequivocal evidence for the involvement of IA at α_1_-AR as an underlying mechanism was not provided.

Other investigators have explored the possible involvement of IA in synaptic transmission and other CNS functions. In rat cardiac vagal neurons, prazosin reduced the frequency of GABAergic and glycinergic neurotransmission, whereas phenylephrine increased it [92]; however, a specific role of inverse agonism in this observation was not proven. In rats with chronic spinal cord injury, presumably lacking endogenous noradrenaline release from fibers originating from the brainstem, α_1_-AR agonists methoxamine and the α_1A_-selective A 61603 facilitated Ca^2+^-mediated persistent inward currents and produced muscle spasms both in vivo and in vitro, whereas in vivo recorded spasms were inhibited by WB 4101, prazosin, and Rec 15/2739 in the absence of agonists [93]. In contrast, only WB 4101 and prazosin blocked spasms in vitro. Transgenic mice expressing CAM α_1A_- but not α_1B_-AR exhibited antidepressant-like behavior in the tail suspension test and forced swim test [96], which was reversed by prazosin and mimicked by the chronic treatment of WT mice with cirazoline. While an effect on CAM indicates a possible involvement of IA, definitive proof was lacking. This problem of proving the involvement of IA in in vivo studies is also highlighted by a study with doxazosin in patients with allergic rhinitis, where doxazosin reduced the peak nasal inspiratory flow, whereas oxymetazoline increased it [94].

In conclusion, data from isolated tissues are highly suggestive of the possibility of observing IA with natively expressed α_1_-AR. While some in vivo data applying known inverse agonists are compatible with this view, the study designs did not allow unequivocal differentiation between IA and the antagonism of endogenous neurotransmitters in most cases.

### 5.2. α_2_-AR

Only a few in vivo reports relate to IA at α_2_-AR, in most cases not providing conclusive evidence that the observed effects are mediated by IA. In cultured human meibomian gland epithelial cells, the α_2_-AR agonists brimonidine and clonidine promoted differentiation and inhibited proliferation, whereas RX 821002 and MK 912 failed to inhibit this and, if anything, acted as partial agonists on differentiation [97]. In mice with syngeneic transplants of mammary duct carcinomas, the α_2_-AR agonists clonidine and dexmedetomidine enhanced tumor growth, which was inhibited by yohimbine and rauwolscine; in the absence of clonidine, rauwolscine reduced tumor growth, with yohimbine having a smaller and inconclusive effect [98].

Others have investigated the effects of α_2_-AR ligands on the inhibition of food intake in rats by sibutramine or bupropion. Imiloxan, atipamezole, BRL 44408, and RX 821002 alone did not affect the food intake at the chosen doses [99]. Atipamezole and RX 821002 increased meal sizes in the presence of sibutramine, and BRL 44408 and imiloxan inhibited it, whereas the other two antagonists did not. The situation became even more complex in a follow-up study where the α_2_-AR antagonists were studied in conjunction with bupropion [100]. In this setting, imiloxan, atipamezole, and BRL 44408 similarly increased the inter-meal interval; however, BRL 44408 reduced meal initiation, whereas imiloxan increased it modestly, and atipamezole increased it markedly. Only imiloxan reduced the size of the first meal. The interpretation of these observations is complicated based upon a comparison of single doses and differential α_2_-AR subtype recognition profiles of the antagonists. In rats with chronic spinal cord injury, clonidine and UK14303 decreased excitatory postsynaptic potentials, whereas RX 821002 increased them [93].

### 5.3. β-AR

#### 5.3.1. Heart

The role of IA has been investigated more often in the heart than in any other tissue. Studies were based on isolated tissues or performed in vivo in healthy animals with endogenously expressed receptors [58,90,101] or humans, including those with naturally occurring polymorphisms of the receptor [102]. They also included data from knock-out models [103] and those with the transfection of receptors [89,103,104] or transgenic (over)expression of WT [47,104,105,106,107,108,109,110,111,112,113,114]. The animal models most often involved mice, but rats [44,58] and rabbits [101,104] were used as well. Other than healthy animals, animal models of disease including coronary heart disease [115] or arrhythmia [116] were studied, or material derived from patients suffering from heart failure [55,102,104,117,118]. Such studies have been performed at various levels of cardiac function and readouts, including signal transduction [89,90,101,103,105,110,112,117], cardiomyocyte electrophysiology [102,107,108,109,113,114], inotropy [47,89,90,103,104,106,110,111,112,113,115,117,118,119], lusitropy [113], chronotropy [44,58], and conduction [116]. Hereafter, the resulting data will be discussed by the level of investigation.

Signal transduction in the heart has mostly been studied as cAMP formation. As the β_2_-selective ICI 118,551 has consistently been reported to be an inverse agonist at β_2_-AR (see above), several investigators reported the lowering of cAMP formation by ICI 118,551 in most studies with cardiac tissue from mice transgenic for the WT receptor [104,105,110,112,114]; β_1_/β_2_-double KO mice transfected with either a β_2_-AR or β_1_/β_2_ chimeric receptor (but not with a β_1_-AR) [89,103]; and, most importantly, also in many [90,110,112], but not all, studies [104,114] with WT mice. One study in healthy rabbits did not detect IA for ICI 118,551, even if cAMP formation had been enhanced by treatment with pertussis toxin [101]; however, that study used a lower concentration of ICI 118,551 than all others (10 nM), which may have been insufficient to elicit a robust response. The lowered cAMP level was indeed linked to IA, as was demonstrated by the observation that alprenolol did not reduce cAMP formation, but blunted the effects of ICI 118,551 [105]. The infusion of WT mice for 14 days with ICI 118,551 increased protein kinase A activity, whereas the infusion of atenolol or carvedilol did not; in contrast, all three ligands reduced protein kinase A activity in transgenic mice overexpressing the WT β_2_-AR [110]. In contrast, G protein receptor kinase 2 was only increased by carvedilol in WT and only sharply reduced by alprenolol and carvedilol. The authors proposed that these regulations were not driven by spontaneous activity of the receptor, but rather by occupancy.

Several investigators have explored spontaneous activity and IA, predominantly by using transgenic mice overexpressing WT β_2_-AR in comparison to control animals. Initial studies reported greater Ca^2+^ sparks in the transgenic mice, which was normalized by ICI 118,551 [113]. In contrast, the properties of the L-type Ca^2+^ channel were found to be unaltered and not affected by ICI 118,551 [114]. While others reported lower activity of the L-type Ca^2+^ channel in transgenic mice, they also failed to observe IA by ICI 118,551 in this model [108]. That group also reported that the activity of the hyperpolarization-activated I_f_ current was markedly enhanced in the transgenic mice and, in contrast to the Ca^2+^ channel activity, shifted towards values observed in WT mice by ICI 118,551 [107]. Based on mRNA measurement, they proposed that this may at least partly occur secondary to an enhanced expression of cyclic nucleotide-gated HCN channels, specifically HCN 4. Modeling and simulation studies based on the data reported by [105] predicted that ICI 118,551 should not affect the voltage of the action potential or the magnitude of the background Ca^2+^ transients in WT mice, but reduce it in transgenic animals [119].

Many investigators have explored possible IA in cardiac contraction and, more rarely, relaxation [113]. Such studies were reported with mouse [47,89,90,103,106,112,114,115], rat [104,111,112], rabbit [104], and human tissue [47,102,104,117,118]. Several of the mouse studies involved transgenic overexpression of the human β_2_-AR [106,110,111,112,114] and more rarely, β_1_-AR [47], and some of the others included transfection with β_2_-AR and/or G_i_ [89,103,104]. Studies mostly focused on the effects of acute exposure to β-AR ligands, but in some cases, also chronic treatment [110]. Some animal [115] and most human studies involved material from diseased subjects [102,117].

In studies with transgenic expression of the β_2_-AR, multiple groups independently reported that ICI 118,551 reduced basal cardiomyocyte contraction [104,105,106,111,112,113,114]. Such inhibition was abolished after the inactivation of G_i_ by pertussis toxin. In line with this, it was also reported that the overexpression of β_2_-AR in rabbit cardiomyocytes or G_i_ in rat cardiomyocytes allowed the detection of IA by ICI 118,551 [104]. Similar IA was also reported with carvedilol [111]. Others have used cardiomyocytes from β_1_/β_2_ double KO mice transfected with β_1_-AR, β_2_-AR, or chimeras thereof [89,103]: Upon transfection with β_2_-AR, the inhibition of basal contraction was observed with ICI 118,551, but not with CGP 20,712 or propranolol; inhibition by ICI was also observed upon transfection with the chimeric receptor, but not with the β_1_-AR. In contrast to an observation by others [104], the IA of ICI 118,551 was insensitive to treatment with pertussis toxin in those studies [89].

While these studies demonstrate that IA is detectable for cardiomyocyte contraction if the basal tone of the system is increased by an enhanced expression of receptor or G proteins, the clinically more relevant question is whether this also applies in the absence of such enhancement of the basal tone. In contrast to the above study with transgenic expression of the β_2_-AR, reductions of the basal cardiomyocyte tone were not detected in cardiomyocytes from WT mice [112,114] or, when detected, were much weaker than in transgenic mice [104,106]. They were also not detected in cardiomyocytes from rats or rabbits in the absence of additional interventions [104]. However, a reduction of basal contractility by ICI 118,551 became detectable in WT mice when the basal tone was increased by the PI-3-kinase inhibitor LY 294,002 or the phosphodiesterase 3/4 inhibitor milrinone [90]. One group of investigators reported negative inotropic effects of various β-AR antagonists, including acebutolol, alprenolol, atenolol, metoprolol, ICI 118,551, nadolol, pindolol, propranolol, and timolol in the heart of reserpinized rats and, investigated in less detail, in those of untreated rats and reserpinized mice [58,120]. In reserpinized rats, the extent of negative inotropic effects varied between the left atrium, right atrium, right ventricle, and right papillary muscle for all compounds. However, two findings question whether these effects are related to IA: Firstly, the concentrations required to observe negative inotropy were >1 µM, and secondly, the extent of negative inotropy did not align with the presence or absence of IA as detected by other investigators (see above), except that it was weakest with the known weak partial agonist pindolol. On the other hand, the negative inotropic effects of atenolol, ICI 118,551, or propranolol were partly inhibited by pindolol. Therefore, it remains unclear whether these findings can be attributed to IA or even to β-AR.

Studies on IA in human hearts have involved failing heart samples, mixed groups of failing, and non-failing hearts, or parallel investigations of both. One group reported that metoprolol, but not carvedilol or bucindolol, reduced isoprenaline-induced contraction in failing human hearts to levels lower than the baseline [117]. In a follow-up study, they reported that both bisoprolol and nebivolol reduced the forskolin-stimulated force of contraction in a ventricular strip of a mixed group of patients with and without heart failure; a concentration-dependent reduced inotropy was also observed for both compounds and also bisoprolol (but not for bucindolol or carvedilol) in atria from non-failing hearts [118]. Accordingly, bucindolol abolished the negative inotropic effect of nebivolol. Other investigators reported a lack of a negative inotropic effect in ventricular human cardiomyocytes under conditions where one was observed in samples from failing hearts [104]. A third group reported on a mixed population of ventricular strips of non-failing and failing human hearts and analyzed contractile responses based on the genotype for a β_1_-AR gene polymorphism [102]. Isoprenaline had greater responses in non-failing than failing hearts, and within each group in homozygous Arg^389^ subjects than in those carrying at least one Gly^389^ allele; in a mixed group of failing and non-failing hearts, carvedilol behaved as a neutral antagonist, whereas bucindolol was neutral in Gly^389^ carriers, but elicited a negative inotropic response in homozygous Arg^389^ subjects.

Applying a very different approach, the group of Bond from Houston, TX, did not acutely add inverse agonists, but rather administered them for multiple days or weeks. In an initial study, alprenolol, carvedilol, ICI 118,551, or propranolol were given to WT and mice with transgenic overexpression of the β_2_-AR for 4 days [111]. ICI 118,551 and propranolol, and to a limited extent and not always reaching statistical significance carvedilol, but not alprenolol, reduced the elevated presence of G_i_ in the heart of the transgenic mice, increased G_s_, and further increased GRK2. In WT mice, neither affected G_i_, and only ICI 118,551 increased G_s_, but ICI 118,551, propranolol, and carvedilol increased GRK2, whereas alprenolol had no effect on any of the three proteins. In line with this pattern of altered protein expression, treatment with ICI 118,551, carvedilol, and propranolol increased the basal tension of isolated atria by 150, 141, and 129 mg compared to 96 mg in non-treated mice; tension in alprenolol-treated animals was 90 mg. The acute addition of ICI 118,551 reduced atrial tension, and this effect was markedly greater than in historical controls [105]. In a follow-up study, WT and transgenic mice were treated for 14 days with ICI 118,551, carvedilol, or alprenolol [110]. Cardiac protein kinase A activity was increased in WT mice treated with ICI 118,551, but not with carvedilol or alprenolol, whereas the increased activity in the β_2_-AR transgenic mice was reduced to WT levels by all three ligands. The increased GRK2 expression levels in transgenic mice was lowered by treatment with alprenolol or carvedilol, but not with ICI 118,551; in contrast, only carvedilol increased GRK2 expression levels in WT mice. The increase in basal atrial tension in transgenic mice upon treatment with ICI 118,551 or carvedilol [111] was not confirmed in the follow-up study, but the depressed inotropic response to histamine in the transgenic mice was partly restored by all three ligands [110]. The authors interpreted these complex findings with an inverse agonist and a neutral antagonist to indicate that it was receptor occupancy and not spontaneous activity driving the changes of GRK and inotropic histamine responses. In another follow-up study, the authors used WT mice that had undergone myocardial infarction followed by 3 weeks of treatment with ICI 118,551, carvedilol, or alprenolol [115]. Myocardial infarction led to a reduced mitral-wave E peak velocity and aortic peak velocity; after 3 weeks of treatment, it was improved by carvedilol to pre-occlusion values, whereas it worsened in the alprenolol and non-treated groups. Neither treatment affected the aortic peak velocity. While permanent occlusion of the left anterior descending coronary artery reduced inotropic responses to isoprenaline in left atria, treatment with carvedilol markedly enhanced it. The authors proposed that the beneficial effects of carvedilol may reflect a combination of IA at β_2_-AR and antagonism at β_1_-AR.

A possible role of IA at β-AR has also been explored for the regulation of chronotropy and cardiac conduction. Mice with the heart-specific transgenic overexpression of β_1_-AR exhibited a greater spontaneous beating right of isolated right atria [47]. The increased heart rate was not affected by reserpinization. However, it was reduced by various β-AR ligands, with a rank order of CGP 20,712 > bisoprolol > metoprolol > carvedilol, whereas carvedilol, if anything, increased it. A reduction of the in vivo heart rate was also observed in bisoprolol-treated mice by an scFv antibody fragment with a high affinity for the β_2_-AR [44]. A different antibody against β_2_-AR induced conduction blocks in the murine heart; however, it remains unclear whether this can be attributed to IA because ICI 118,551 did not enhance, but rather reverse, this [116].

#### 5.3.2. Lung

Although studied to a lesser extent than in the heart, IA at β-AR has also been explored in the airways, largely by two groups of investigators. The Bond group from Houston, TX, initially showed that the treatment of mice with nadolol or ICI 118,551, but not with metoprolol, had a protective effect on the ovalbumin-induced airway hypersensitivity model of asthma [121]. This was associated with an upregulation of β_2_-AR and a reduced expression of several proteins involved in the regulation of bronchial tone, including G_i_, phosphodiesterase 4D, and phospholipase C-β1. However, these findings did not allow a clear conclusion on the possible involvement of IA in the observed effects of ICI 118,551 and nadolol. To obtain further evidence, the asthma model was applied to WT and β_2_-AR knock-out mice [122]. The knock-out mice exhibited a similar phenotype to those treated with an inverse agonist, including a reduced airway hyperresponsiveness. In contrast, treatment with alprenolol did not mimic the effects of the inverse agonists, providing additional evidence that the anti-asthmatic effects in the mouse model may be explained by IA. They also reported that the co-administration of nadolol with dexamethasone was more effective in the murine ovalbumine model than either drug alone [123]. Finally, they used HEK923 cells transfected with human β_2_-AR to directly compare the acute and chronic effects of nadolol [26]. Basal cAMP levels were reduced by the acute (5 min) addition of nadolol, but increased by prolonged exposure (24 h). Similarly, acute nadolol decreased forskolin-stimulated phosphorylation at the β_2_-AR protein kinase A site Ser^262^, whereas prolonged exposure increased it. Chronic exposure to nadolol also increased β_2_-AR protein levels and decreased receptor degradation, consistent with receptor stabilization by the inverse agonist; it also increased cellular levels of G_αs_, but had no effect on G_αi_. Overall, these observations in HEK293 cells were consistent with those in the murine asthma model and lend additional support to the involvement of IA in the observed bronchoprotective effects.

The group of Zaagsma and Meurs from Groningen, The Netherlands, used bovine tracheal smooth muscle strips and found that an 18 h treatment with fenoterol reduced airway contractility; when various antagonists were added after the fenoterol treatment (all dosed to achieve 98–99% receptor occupancy), they restored airway contractility with a rank order of efficacy of pindolol ≥ timolol = propranolol > alprenolol ≥ sotalol > labetalol [124]. Of note, these effects do not necessarily imply IA as the underlying mechanism, particularly as the observed rank order does not match that observed for the extent of IA (see above). In a follow-up study, they confirmed the original findings with fenoterol; timolol added after pre-treatment with fenoterol restored airway contraction, but it remains unclear whether IA was involved, as timolol had no effects on airway contraction in the absence of pre-treatment with fenoterol [125]. Other investigators found that salbutamol increased the short circuit (I_sc_) current in Calu-3 human airway adenocarcinoma cells, whereas carvedilol decreased under basal conditions and after stimulation with a cAMP-mimetic [126]. The carvedilol effect was abolished after pretreatment with ICI-118551, which questions whether IA was involved in this observation.

Inflammatory cells play important roles in the pathophysiology of obstructive airway disease. IgE-dependent activation of human mast cells opened the intermediate conductance Ca^2+^-activated K^+^ channel iK_Ca_1 to cause hyperpolarization and enhancement of both Ca^2+^ influx and degranulation. While salbutamol inhibited iK_Ca_1 currents, ICI 118,551 opened such channels [37]. In antigen-specific T lymphocyte lines, (R)-albuterol inhibited proliferation, whereas (S)-albuterol did not [127]. However, the addition of (S)-albuterol to (R)-albuterol concentration-dependently increased proliferation and both the inhibitory effects of (R)-albuterol alone and the stimulating influence of (R)- plus (S)-albuterol were inhibited by propranolol. The authors hypothesized that the (S)-albuterol may behave as an inverse agonist at T-cell β_2_-AR, but definitive evidence was not provided.

Unrelated to airways or β_2_-AR, it was reported that CGP 20,712 or the combination of CGP 20,712 with ICI 118,551 (but not ICI 118,551) reduced cAMP formation in rat pituitary cell aggregates, whereas either compound alone inhibited the stimulatory effects of isoprenaline [81]. The inhibitory effects of CGP 20,712 were abolished by pertussis toxin or carvedilol, but not by propranolol or betaxolol, with none of these three ligands affecting cAMP accumulation in the absence of CGP 20,712. While the authors attributed the inhibition by CGP 20,712 to IA, its modification pattern by other β-AR ligands and the lack of effect of ICI 118,551, carvedilol, betaxolol, or propranolol do not support this interpretation.

## 6. Molecular Mechanisms and Structural Basis of Inverse Agonism at AR

Advances in the structural exploration of receptor conformations and ligand-induced changes thereof since the mid-2000s have enabled an improved understanding of inverse agonism and its relationship to agonism. Much of this work has been done using the β_2_-AR as a model system. The first crystal structures were obtained by locking the receptor into an immobile state, supported by binding to the inverse agonist carazolol [128,129]. Structure determinations suggest that IA-induced changes are ‘different’ [130] or ‘opposite’ [78,131,132] to agonist-induced changes. This view is supported by molecular dynamic simulations [133]. Structural modeling suggests that a key change in the receptor conformation is induced by different degrees of tilting of the helix V [134] and, more specifically, an outward movement of the helices V and VI coupled with inward movements of the 3rd and 7th helix [135]. Another structural indicator could be a methionine residue at position 82 [136]. An IA-induced receptor conformation appears to be different from an empty structure, which may not be straightforward to conceptualize considering that both conformations display a similarly low activity level [137].

Crystallization data also enables computational prediction of the degree of (inverse) agonism [138,139]. It has been shown that virtual screening processes can be set up to preferentially deliver either agonists [140] or inverse agonists [141].

NMR spectroscopy has been reported to map three [142], and molecular dynamic simulations to map seven receptor activation states to different energy states [143]. Some ligands (‘true’ neutral antagonists) do not appear to shift the receptor activity level set by specific assay conditions [144]. Different types of ligands (agonists, antagonists, and inverse agonists) modulated the conformational dynamics differently [145]. IAs have been shown to induce immobile receptor structures (whereas agonists induce more mobile, dynamically fluctuating conformations) [146]. These more immobile structures cannot readily bind G proteins [147]; however, these conformations may be able to activate different signaling pathways [148].

At β_2_-ARs, different Gs protein species select for different receptor activity levels, thereby leading to a different potential for action, and action, of inverse agonists [32,84]. Moreover, mutations in the receptor can induce constitutive activity, with an increase in precoupling potentially playing a role in this [46]. The activity level of constitutively active mutants can be restored to more ‘normal’ levels by inverse agonists [30,80]. Similar results have been shown for constitutive recycling, which could be inhibited by IAs [80]. However, some ligands do not change the receptor confirmation, as measured by FRET for both WT and CAM α_2_-ARs [68]—this may be at odds with the idea that IAs reduce specific receptor confirmations with specific affinities for G proteins.

Receptor expression levels can change the apparent properties of ligands (inverse vs. partial agonism) [67]. Conversely, the antagonist betaxolol and the inverse agonist phentolamine differently regulate the expression levels of wild-type and constitutively active β_2_-ARs [27]. Inverse agonist [15] and general signaling [14] properties of a ligand differ between different downstream signaling pathways (e.g., G protein vs. G protein-independent), or G_s_ vs. G_i_ signaling [31]. Similar implications are considered by examining lipid rafts, which can help stabilize receptors in an inactive state [24].

Agonists and inverse agonists may be able to stabilize different multimolecular complexes, e.g., receptor monomers vs. dimers [149]. G proteins—presumably by shifting the average conformations towards more active states—reduce the number of binding sites for inverse agonists [88] and could thus be considered as allosteric modulators [150]. Moreover, different stoichiometries of Gα, Gβγ, or guanosine nucleotides can also change the receptor activity levels [151]. Inverse agonists have been reported to not change the association of receptors and G proteins [29], which could be explained by a high degree of basal precoupling.

## 7. Conclusions and Implications for Drug Development

The degree of investigation of IA at AR subtypes differs markedly, particularly at the level of endogenously expressed receptors at the tissue level, with β_2_-AR having been studied the most frequently and b_3_-AR apparently not having been studied at all. This has led to a heterogenous robustness of the existing evidence, but some common themes have emerged. The detected IA at a given subtype and model system varies between compounds, but some chemical compound classes may intrinsically exhibit greater IA than others. For instance, quinazolines appear to exhibit a greater degree of IA at α_1A_- and α_1B_-AR than *N*-arylpiperazines (Figure 3). However, minor chemical modifications may lead to major changes of the degree of IA within a compound class [33].

IA is best observed when the tone of the system is increased, for instance, by the overexpression of the receptor, presence of a CAM, or increased expression of the coupled G protein (Figure 2). Accordingly, IA can also be observed with endogenously expressed receptors, but this is a less consistent finding than in settings with an increased tone of the system; when IA is detected with endogenously expressed receptors, it is typically of a smaller magnitude than in settings with an increased tone, as is also observed for receptors other than AR [8]. Of note, the increased expression of a receptor and/or a specific G protein can occur in the context of various physiological and pathophysiological settings [152] or be caused by the administration of drugs that affect gene transcription and/or stability of the expressed protein, for instance, glucocorticoids. An example of physiological differences in tone is the presence of β_1_-AR gene polymorphism [55,102]. As many elements of the cardiac β-AR signal transduction cascade exhibit an altered expression in congestive heart failure and many clinically used β-AR antagonists exhibit various degrees of IA, it has been speculated that the presence of IA may at least partly explain different outcomes upon the treatment of heart failure patients with different β-AR antagonists [153]. However, clear conclusions on the role of IA are hampered by the fact that clinical outcomes most likely do not only depend on IA, but also other properties of the various drugs, including selectivity for β_1_- over β_2_-AR and biased agonism. Another pathophysiological example is obstructive airway disease, particularly asthma. While the use of β-AR antagonists is classically considered contra-indicated in asthma patients, it has been shown that slow up-titration is well-tolerated in asthmatic patients and may even have beneficial effects [154]. As β-AR antagonists with different degrees of IA had differential profiles in an animal model of asthma [155], it has been proposed that a combination of the choice of compounds with strong IA and slow up-titration of their dosages may lead to the effective and innovative management of asthma patients [156]. However, others have remained skeptical about this possibility [157].

Based on all of the above, the question emerges of whether IA should play a role in drug discovery and provide a development perspective, e.g., as part of a target product profile for new compounds. Whether IA is a potential therapeutic strategy or just a pharmacological curiosity has been debated since the early days of IA research [7,158]. A potential advantage of such considerations would be that differences in tone of a given AR or GPCR in general—between cell types and tissues, and between healthy and diseased subjects—could be leveraged to obtain functional selectivity, i.e., a good effectiveness with limited adverse events. A potential disadvantage of this is that we typically lack an in-depth knowledge of the tone in the various cell types and tissues important in a certain pathophysiology, not to mention possible alterations of that tone in disease; this makes it hard to predict what the optimal compound should look like. An added layer of complexity is that IA is certainly not the only drug property to be considered for the definition of a target product profile and others, such as biased agonism [159], ortho- vs. allosteric receptor modulation, subtype-selectivity, and pharmacokinetics, also weigh in. These considerations are similar to those using biased agonism as a desired drug property in the target product profile, where we have recently argued that it may be too early to define biased agonism as a desirable drug property, particularly for highly innovative treatments where prior knowledge on cell types and signaling pathways involved and their endogenous role in disease is limited [160]. However, others have argued to the opposite [161]. A similar debate is expected for the role of IA in drug discovery and development and only future experience will teach us when and where IA will be a differentiating factor for new drugs.

## Figures and Tables

**Figure 1 cells-09-01923-f001:**
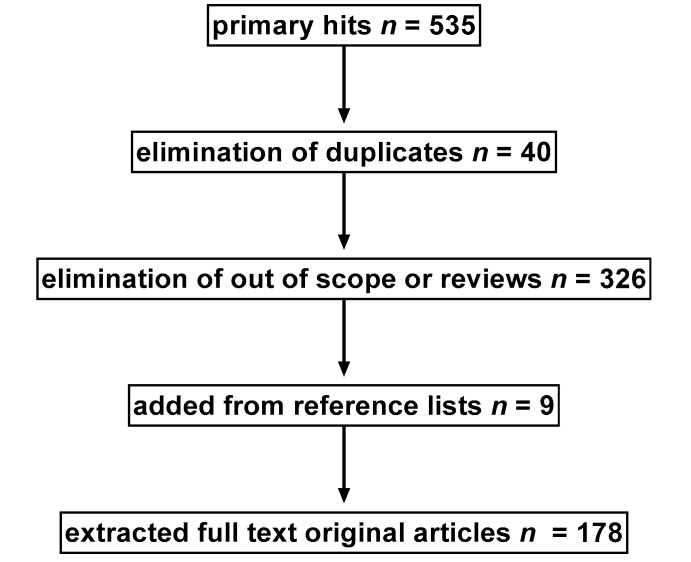
PRISMA flow chart of the handling of search results.

**Figure 2 cells-09-01923-f002:**
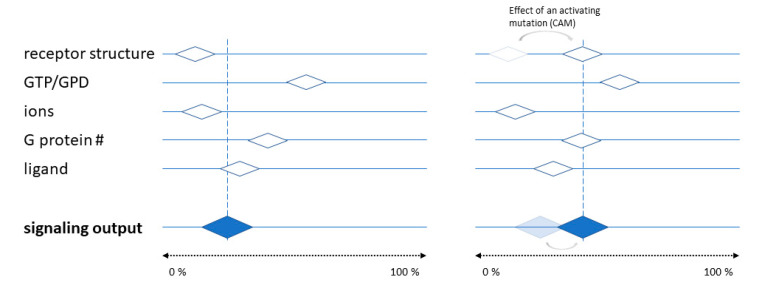
Ligand efficacy as one of many determinants of the signaling output. A number of effects determine the signaling output (e.g., the receptor structure, GTP/GDP ratio, ion concentrations, number of G proteins, and ligand binding) (**left**). The apparent property of a ligand as an inverse agonist, neutral antagonist, and agonist can only be determined relative to the overall output level. For example, introducing a constitutively active mutant (CAM) into the receptor structure may change a ligand’s effect from weak partial agonism to inverse agonism (**right**).

**Figure 3 cells-09-01923-f003:**
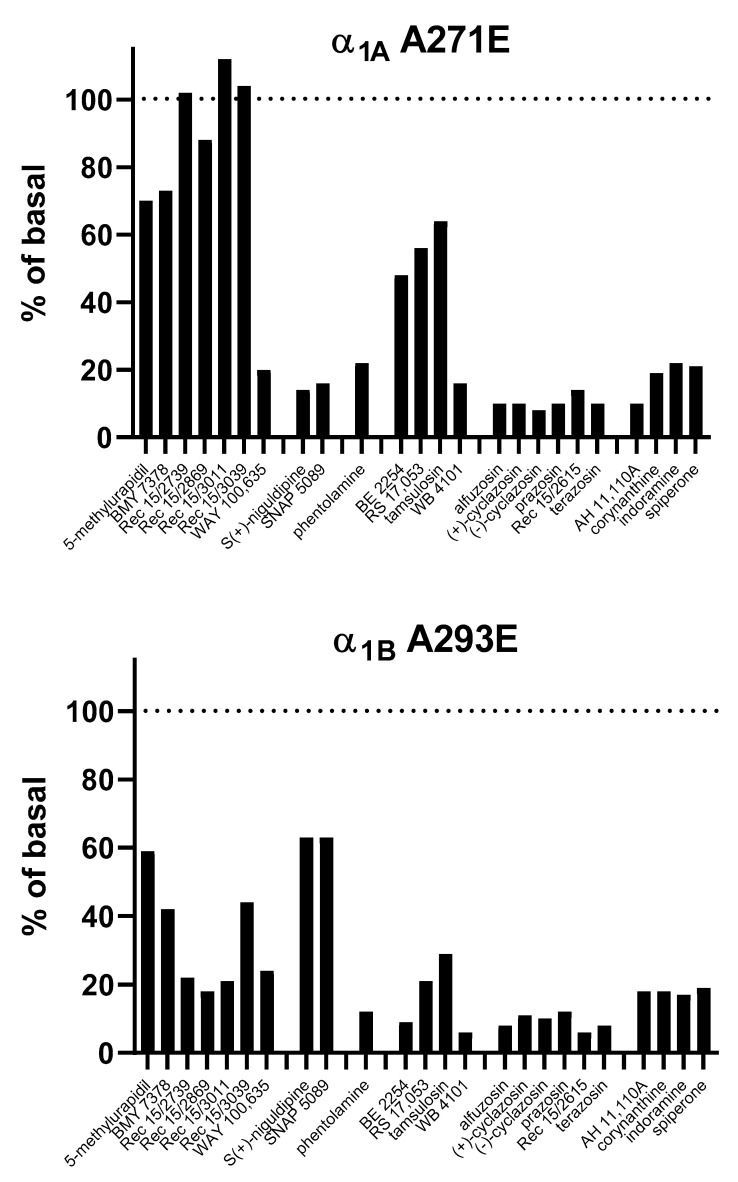
Inhibition of basal inositol phosphate formation in cells stably transfected with CAM of α_1A_-adrenoceptors (AR) (A271E mutation) and α_1B_-AR (A293E mutation). The figure was modified with permission from [19]. All compounds were tested at a concentration of 10 µM, except for Rec 15/3039 (100 µM), and data are shown as means of three–six experiments. Compounds are grouped based on structural similarities as classified by the original authors from left to right as *N*-arylpiperazines, 1,4-dihydropyridines, imidazolines, benzodioxans and phenylalkylamines, quinazolines, and ‘other’.

**Figure 4 cells-09-01923-f004:**
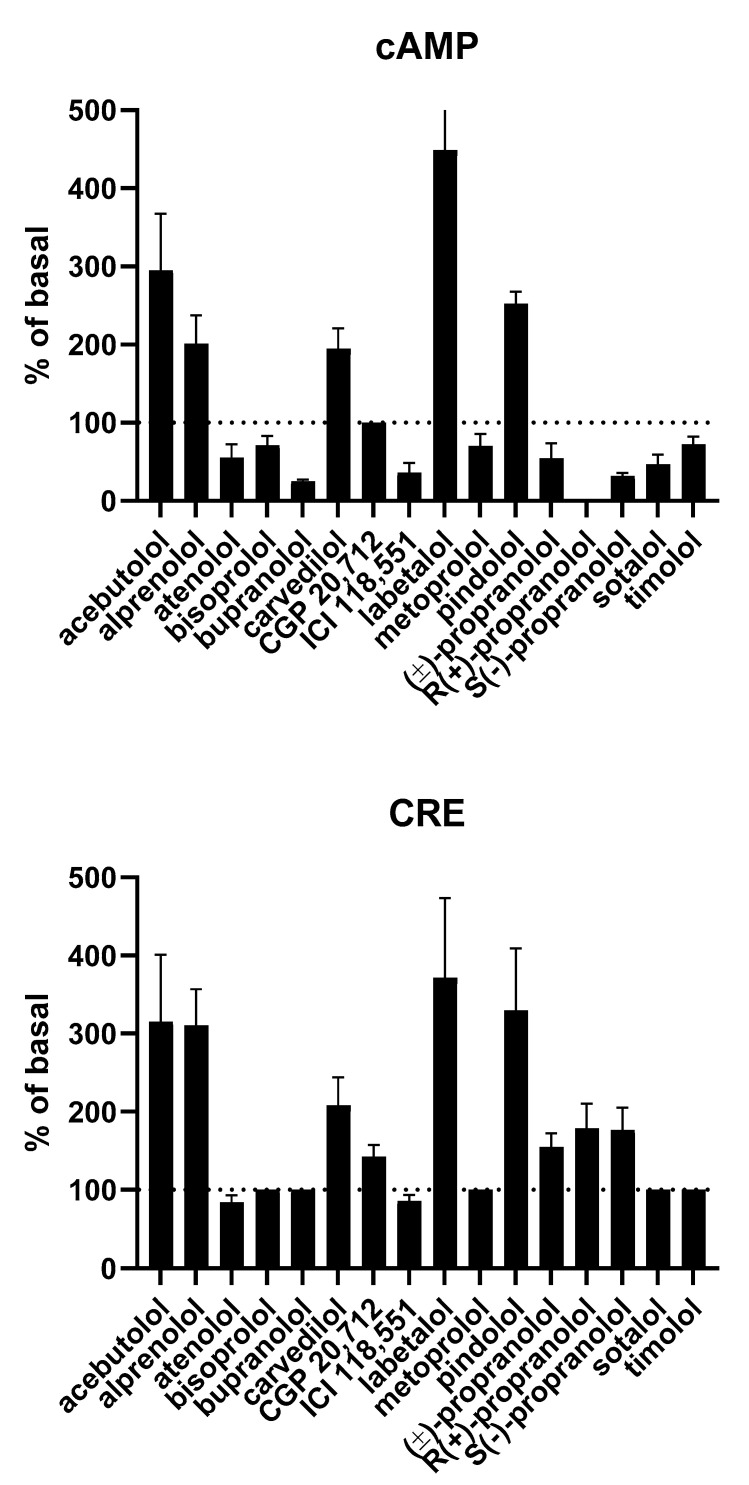
Efficacy of ligands for cAMP accumulation and the activation of a cAMP response element (CRE) reporter gene in cells stably transfected with human β_2_-AR. The figure was generated based upon data from [15]. Each bar represents means ± SD of E_max_ as the % of basal derived from 3–38 concentration-response experiments. Note that the efficacy of the most efficacious ligand, labetalol, was 8.5% and 66.9% of the maximum isoprenaline response in the cAMP and CRE assay, respectively.

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
