# Peer review of "A Systematic Review of Inverse Agonism at Adrenoceptor Subtypes"

_cells, 2020, doi:10.3390/cells9091923_

Round 1

Reviewer 1 Report

The manuscript by Michel, Michel-Reher and Hein is a well written comprehensive review of the current status of inverse agonism at adrenoceptor subtypes, a topic that has not been covered extensively in recent reviews.  This topic will be of great interest to the readership. 

Minor concern

IN abstract, the last but one sentence beginning….The extent ……by chemical class is not clear.  Please correct the sentence

Author Response

The manuscript by Michel, Michel-Reher and Hein is a well written comprehensive review of the current status of inverse agonism at adrenoceptor subtypes, a topic that has not been covered extensively in recent reviews.  This topic will be of great interest to the readership. 

Minor concern

IN abstract, the last but one sentence beginning….The extent ……by chemical class is not clear.  Please correct the sentence

Reply: Done (l. 23-24).

Reviewer 2 Report

This is an interesting and valuable review on inverse agonism (IA) at adrenoceptor (AR) subtypes. This review includes a systematic description of many compounds exhibiting IA at AR subtypes. This information is valuable but hard to be interpreted only from the text. Providing tables summarizing information of the compounds cited would help readers to follow.

Following points are needed to be addressed.

1) Lines 83–84: The relation between IA at α2-adrenoceptors and GTPγS binding should be explained.

2) Line 107 “partial IA”: It needs to be explained.

3) Lines 117–120: This sentence is difficult to be interpreted. More detailed explanation utilizing figures would help readers to follow.

4) Line 235 “and of noradrenaline”: This may mislead readers.

5) Lines 298–299: This sentence should be checked.

6) Line 302 “this was interpreted as evidence for precoupling”: A more detailed explanation should be included.

7) Line 304 “transfected with both WT and CAM receptor”: Correct?

8) Lines 321–322: This sentence should be checked.

9) Line 327 “transfected PC12 cells”: Show what was transfected.

10) Figure 4 “labetolol”: “labetalol”?

11) Line 437 “patter”: “pattern”?

12) Lines 439–440 “detected for cAMP formation but not the reporter gene construct for some compounds”: In reference #15, CRE-gene transcription is suggested to be stimulated through the pathway independent of cAMP. More explanation should be included so as not to mislead readers.

13) Line 447 “change of Gs protein or mRNA”: What changed in Gs protein or mRNA?

14) Lines 477–478: This sentence should be checked.

15) Lines 514–515: This sentence should be checked.

16) Lines 541–543: What is the exception? Please explain more detail.

17) Lines 549–550: Which reference is this description about? Which is the effect of carvedirol, increasing or reducing?

18) Line 559 “values”: What are the values?

19) Lines 639–641: This sentence needs more detailed explanation. How is histamine response involved?

20) Lines 653–654 “A reduction of in vivo heart”: What is the reduction?

21) Lines 738–739: This sentence should be checked.

21) Lines 753–754: This sentence should be checked.

Author Response

General reply: We have tried to address the specific points raised by you but experienced a major problem: the line numbers you mentioned in most cases do not match any wording you mention, indicating that something went wrong between the manuscript version we had submitted and the one you have reviewed. By searching for the wording you cited we tried our best to identify which section you had in mind, but in some cases where you only mentioned line numbers this was unsuccessful. As line numbers have changed due to the modifications made upon request by you or referee 1, we would like to note that all specifically mentioned line numbers hereunder relate to the originally submitted manuscript version and new line numbers are given for comparison.

This is an interesting and valuable review on inverse agonism (IA) at adrenoceptor (AR) subtypes. This review includes a systematic description of many compounds exhibiting IA at AR subtypes. This information is valuable but hard to be interpreted only from the text. Providing tables summarizing information of the compounds cited would help readers to follow.

Reply: We already had considered generating summarizing tables in the original writing process and considered several options. The key reason we decided against this is that for most compound the number of available studies are too small and the different types of assays being reported are too heterogeneous to allow robust conclusions on individual compounds. We feel that a table makes data look more definitive than free text that allows for more grey shading. Rather, other than cataloging the available data based on our systematic review, we wished to focus on the concept that the extent and detection of IA depends not only on the ligand under investigation but also on the context under which it is studied (e.g. cell and tissue type, response or disease state). We tried to conceptualize this in Figure 2. Moreover, we hope that the modified wording on new l. 118-120 makes this clearer. Therefore, it had been an active decision on our part not to generate summarizing tables and rather to show graphs highlighting specific concepts and concomitantly show data on many compounds within one study to enable direct comparisons.

Following points are needed to be addressed.

  • Lines 83–84: The relation between IA at α2-adrenoceptors and GTPγS binding should be explained.

Reply: Unfortunately, we did not understand this comment because l. 83-84 belong to a section on general methodological aspects of how to study IA and do not specifically relate to a2-AR. As described there, GTPγS reflects the direct interaction between ligand, receptor and G protein, thereby being the most simplistic of all frequently applied assays to detect IA and the most proximal in the signal transduction chain.

  • Line 107 “partial IA”: It needs to be explained.

Reply: Unfortunately, we also did not understand this comment because l. 107 or any other line within that paragraph does not talk about partial IA. The first time within the manuscript we talk about partial agonism is old and new l. 38-39 where we explain what is meant by that. Additional explanation in this regard is given in old and new l. 45-54. If these do not address what you had in mind, please explain more specifically what you are concerned about.

  • Lines 117–120: This sentence is difficult to be interpreted. More detailed explanation utilizing figures would help readers to follow.

Reply: We have reworded this sentence  and split into two for greater clarity (new l. 118-120).

  • Line 235 “and of noradrenaline”: This may mislead readers.

Reply: We did not find any mentioning of noradrenaline in l. 235 or the surrounding paragraph. Therefore, we assume that the comment related to old l. 264 (new l. 266). That sentence has been reworded.

  • Lines 298–299: This sentence should be checked.

Reply: The sentence probably was confusing because it included a typo changing the meaning. It should have read “decreased” rather than “increased” related to rauwolscine. This has been corrected (new l. 300).

  • Line 302 “this was interpreted as evidence for precoupling”: A more detailed explanation should be included.

Reply: We did not find any mentioning of precoupling in l. 302 or the surrounding paragraph. Therefore, we assume that the commented related to l. 341. We have provided additional information for clarity.

  • Line 304 “transfected with both WT and CAM receptor”: Correct?

Reply: We did not find such wording in l. 304 and assume that you meant old l. 342. We have reworded this for greater clarity (new l. 340-343).

  • Lines 321–322: This sentence should be checked.

Reply: We have reworded this sentence for clarity (new l. 325).

  • Line 327 “transfected PC12 cells”: Show what was transfected.

Reply: We did not find such wording in l. 327 and assume that you meant l. 368. That sentence was clarified (new l. 370-372).

  • Figure 4 “labetolol”: “labetalol”?

Reply: Thanks for spotting this typo. The figure was corrected.

  • Line 437 “patter”: “pattern”?

Reply: Indeed another typo that has been fixed (old l. 492, new l. 494).

  • Lines 439–440 “detected for cAMP formation but not the reporter gene construct for some compounds”: In reference #15, CRE-gene transcription is suggested to be stimulated through the pathway independent of cAMP. More explanation should be included so as not to mislead readers.

Reply: We did not find such wording in l. 439-440 and assume that you meant l. 494-495. Information on the difference between the cAMP and CRE response in that model has been added to the manuscript (new l. 497-499).

  • Line 447 “change of Gs protein or mRNA”: What changed in Gs protein or mRNA?

Reply: We did not find such wording in l. 447 and assume that you meant l. 503. We have reworded that section (new l. 505-508).

  • Lines 477–478: This sentence should be checked.

Reply: L. 477-478 are part of the legend of figure 4. We have slightly reworded this for clarity (new l. 479). If you had a different part of the manuscript in mind, please explain.

  • Lines 514–515: This sentence should be checked.

Reply: We have reworded that section (new l. 519-521). If you had a different part of the manuscript in mind, please explain.

  • Lines 541–543: What is the exception? Please explain more detail.

Reply: We did not find a mentioning of an “exception” in l. 541-543. This word only is used in l. 45, 151 and 611. We assume that you meant the latter and have reworded it for clarity (new l. 616-617).

  • Lines 549–550: Which reference is this description about? Which is the effect of carvedirol, increasing or reducing?

Reply: The paragraph including l. 549-550 does not deal with carvedilol or any other aspect of b-AR but rather with a1-AR. Since carvedilol is mentioned 34 times within the manuscript, we were unable to guess what you had in mind. We’ll be happy to address this if you explain which specific section you were thinking of.

  • Line 559 “values”: What are the values?

Reply: We did not find a mentioning of an “values” in l. 559 but only in old l. 320, 629 and 718. Obviously, you were not thinking of l. 320. Whether you meant l. 629 or 718 was unclear to us; in both cases the important point is the inverse agonist made values in the transgenic model (old l. 629, new l. 635) or in mice with myocardial infarction (old l. 718, new l. 723) approximate those in wild-type and healthy mice, respectively. If you wish us to clarify further, please let us know which specific passage you are talking about.

  • Lines 639–641: This sentence needs more detailed explanation. How is histamine response involved?

Reply: L. 639-641 and the surrounding paragraph do not include any mentioning of histamine. Do you mean l. 712 or 715? In l. 712 we explicitly stated that we were talking about inotropic responses. Therefore, we feel that it should be clear that old l. 715 (new l. 721) also related to inotropic responses. To be on the safe side, we have added “inotropic” there as well.

20) Lines 653–654 “A reduction of in vivo heart”: What is the reduction?

Reply: L. 653 included information on PTX, l. 654 was an empty line between paragraphs. Did you mean l. 729-730? We have added the apparently missing word “rate” to now clearly read “heart rate” (new l. 736).

21) Lines 738–739: This sentence should be checked.

Reply: We looked at these lines but not find anything to be unclear. Perhaps you had a different section in mind.

21) Lines 753–754: This sentence should be checked.

Reply: We looked at these lines but not find anything to be unclear (except one space too many). Perhaps you had a different section in mind.